# Nonlinear association between C-reactive protein and severity of diabetic foot infection in patients with diabetes: A retrospective cohort study with clinical implications

Li Zhang[1¤a‡], Xufeng Gao[2¤b‡], Meifang He[1¤a], Wenyan Wang[1¤a], Yuebin Zhao[1¤a*]

**1** Department of Endocrinology, Taiyuan City Central Hospital, Taiyuan, Shanxi, China, **2** Department of General Surgery, Children's Hospital of Shanxi, Taiyuan, Shanxi, China

‡ LZ and XG also contributed equally to this work.
¤a Current Address: Taiyuan City Central Hospital, Taiyuan, Shanxi, China
¤b Current Address: Children's Hospital of Shanxi, Taiyuan, Shanxi, China
* 15034186798@163.com

## Abstract

### Objective

Diabetic foot infection(DFI) is a frequent complication of diabetes and can lead to significant morbidity and mortality. These are huge economic health burdens for patients, countries, and the world. Timely diagnosis and accurate assessment of the severity of DFI based on sensitive inflammatory biomarkers is critical, with considerable benefits for debridement procedures and antibiotic use. This study aimed to examine the association between C-reactive protein (CRP) levels and the severity of DFI in patients with diabetes.

### Methods

In this retrospective cohort study, we investigate the association between CRP levels and the severity of DFI in patients with diabetes. Multivariable logistic regression modeling, a generalized additive model, and a two-piecewise linear regression model were conducted to explore the relationship between CRP levels and DFI severity.

### Results

Based on inclusion exclusion criteria, A total of 301 patients with Wagner stage 3 or higher diabetic foot ulcers combined with moderate to severe infections were included in the study at last. CRP levels were independently associated with the severity of DFIs after accounting for potential confounding factors. A nonlinear relationship was observed between CRP level and DFI severity, with a threshold of approximately 105 mg/L. The odds ratio for severe diabetic foot infection was 1.039 (95% confidence interval: 1.019–1.06, P < 0.001) in patients with DFI and CRP < 105 mg/L.

**Data availability statement:** All relevant data are within the manuscript and its Supporting information files.

**Funding:** Taiyuan Bureau of Science and Technology, Science, Technology, and Innovation Program of National Regional Medical Center (202210).

**Competing interests:** The authors have declared that no competing interests exist.

## Conclusion

The relationship between CRP and the severity of DFI was nonlinear; as CRP levels remained below 105 mg/L, the severity of DFI increased with an increase in CRP.

## Introduction

In 2021, the International Diabetes Federation(IDF) notified that over 536.6 million adults global, accounting for more than 10.5% of the global adult population, are expected to have diabetes mellitus, increasing to 12.2% (783.2 million) in 2045 [1]. Diabetes complications continue to increase worldwide, and foot ulcers are one of the most common diabetic complications. Patients with diabetic foot are at high risk of amputation, disability and death when co-infections occur [2,3]. Neuropathy and vasculopathy underlie the pathology of diabetes-related foot ulcers. Complicated infections in addition to diabetes-related foot ulcers can significantly increase serious risks such as amputation and even death [4,5]. These are huge economic health burdens for patients, countries, and the world.

The International Working Group on the Diabetic Foot (IWGDF) guidelines provide recommendations for categorizing diabetic foot infections based on their severity. According to the IDSA and IWGDF criteria, severe diabetic foot infections must have localized infections deeper than the epidermis and subcutaneous tissue, as well as systemic inflammatory response syndrome (SIRS) symptoms [6]. When the Laboratory Risk Indicator for Necrotizing Fasciitis-based (LRINEC) score was applied to diabetic foot infections, it revealed a strong association with amputation and mortality [7]. The score is not specifically designed for diabetic foot infections, and its predictive value takes into account the inclusion of inflammatory markers such as C-reactive protein (CRP) and white blood cell count (WBC). However, the IWGDF guidelines only consider white blood cell counts greater than 12,000 or less than 4,000 cells/mm³ as inflammatory markers, which are also included in the SIRS criteria. Several studies have demonstrated that patients with severe diabetic foot infections (DFI) exhibit significantly higher erythrocyte sedimentation rates (ESR) and C-reactive protein (CRP) levels, as well as significantly lower albumin levels [8–10]. Timely diagnosis and accurate assessment of the severity of DFI based on sensitive inflammatory biomarkers is critical, with considerable benefits for debridement procedures and antibiotic use to decrease the risk of amputation. However, whether the specific relationship between CRP and diabetic foot infection is linear or non-linear is not detailed in any of the current studies. The aim of this study was to investigate the relationship between CRP and the severity of diabetic foot infections (DFI) with Wagner 3 and above ulcers, in particular to explore the non-linear relationship between CRP and DFI severity.

## Materials and methods

### Study design and participants

In this retrospective cohort study, we reviewed the medical records of patients with diabetic foot infections at Taiyuan Central Hospital from January 2020 to September

2023.During the study period, 406 patients were diagnosed with diabetic foot. After excluding patients with Wagner stage 2 and below, cancer and patients with incomplete data. And because the study population was patients with diabetes-related foot ulcers, other inflammatory causes of foot breakdown such as Buerger's disease, inflammatory bowel disease, and rheumatic diseases in diabetic patients were also excluded. A total of 301 patients with Wagner stage 3 or higher diabetes-related foot ulcers combined with moderate to severe infections were included in the research (Fig 1).

This was a cross-sectional study and based on the access to the incidence of diabetic foot infection was 15% [11] with a tolerance error of 0.03, a two-sided test was set up with a = 0.05, and the sample size was calculated to be196.Considering the 20% loss to follow-up rate, a minimum of 236 patients with diabetic foot infection would need to be investigated.

## Covariates

The following potential covariates were analysed in this study, including sex, age, body mass index (BMI) and smoking status, mean arterial pressure, hypertension, coronary heart disease, cerebrovascular disease, fasting plasma glucose, glycosylated haemoglobin, erythrocyte count, haemoglobin, white blood cell count, neutrophil percentage, platelet count, C-reactive protein (CRP), albumin, duration of diabetes, DFI microbial community, severity of diabetic foot infection, amputation status. Smoking status was categorized as current, former, or never smoker [12]. The BMI is the ratio of height to weight squared. Comorbidities(Hypertension, coronary heart disease, and cerebrovascular disease)was determined by documenting in the electronic medical record whether or not a physician has made a previous diagnosis. The DFI microbial community is the result of general bacterial cultures of localised tissue from affected foot ulcers. Amputations were classified as minor or major. Minor amputations were defined as those where the ankle was left intact, and major amputations were above the ankle [13].

## Identification of DFI severity

The International Working Group on the Diabetic Foot has expanded the criteria for diagnosing and evaluating the severity of DFI [14]. 1) Mild infection: Diabetic foot ulcer with no systemic manifestations-involving only the skin or subcutaneous tissue (not any deeper tissues) or any erythema that does not extend > 2 cm. 2) Moderate infection:The ulcers deep to bone and fascia with erythema extending > 2 cm and without systemic manifestations.3) Severe infection: The presence of two or more systemic symptoms, such as body temperature exceeding 38°C or falling below 36°C, a heart rate surpassing 90 beats/min, a respiratory rate higher than 20 breaths/min, a partial pressure $CO_2$ level below 4.3 kPa (32 mmHg), a WBC count exceeding 12,000 or falling below 4,000 cells/mm$^3$, or the presence of more than 10% immature (banded) leukocytes.

## Ethics statement

Approval for this research was granted by the Ethics Review Committee of Taiyuan central hospital (2023046), ensuring full compliance with the ethical guidelines set forth by the Declaration of Helsinki and its successive updates. The requirement for informed consent was waived because the patient data were anonymized.

## Statistical analysis

This study aimed to investigate the association between CRP and DFI severity. There is currently no established diagnostic threshold for C-reactive protein in clinical practice. Patients in this study were divided into three groups based on CRP quantiles to ensure a relatively balanced population between groups and to explore some new findings. Data are presented as mean ± standard deviation (SD) or median (interquartile range, IQR) for continuous variables, and as frequencies (percentages) for categorical variables. Logistic regression analyses was used to estimate the odds ratio (OR) and corresponding 95% CI for DFI severity according to serum CRP levels.

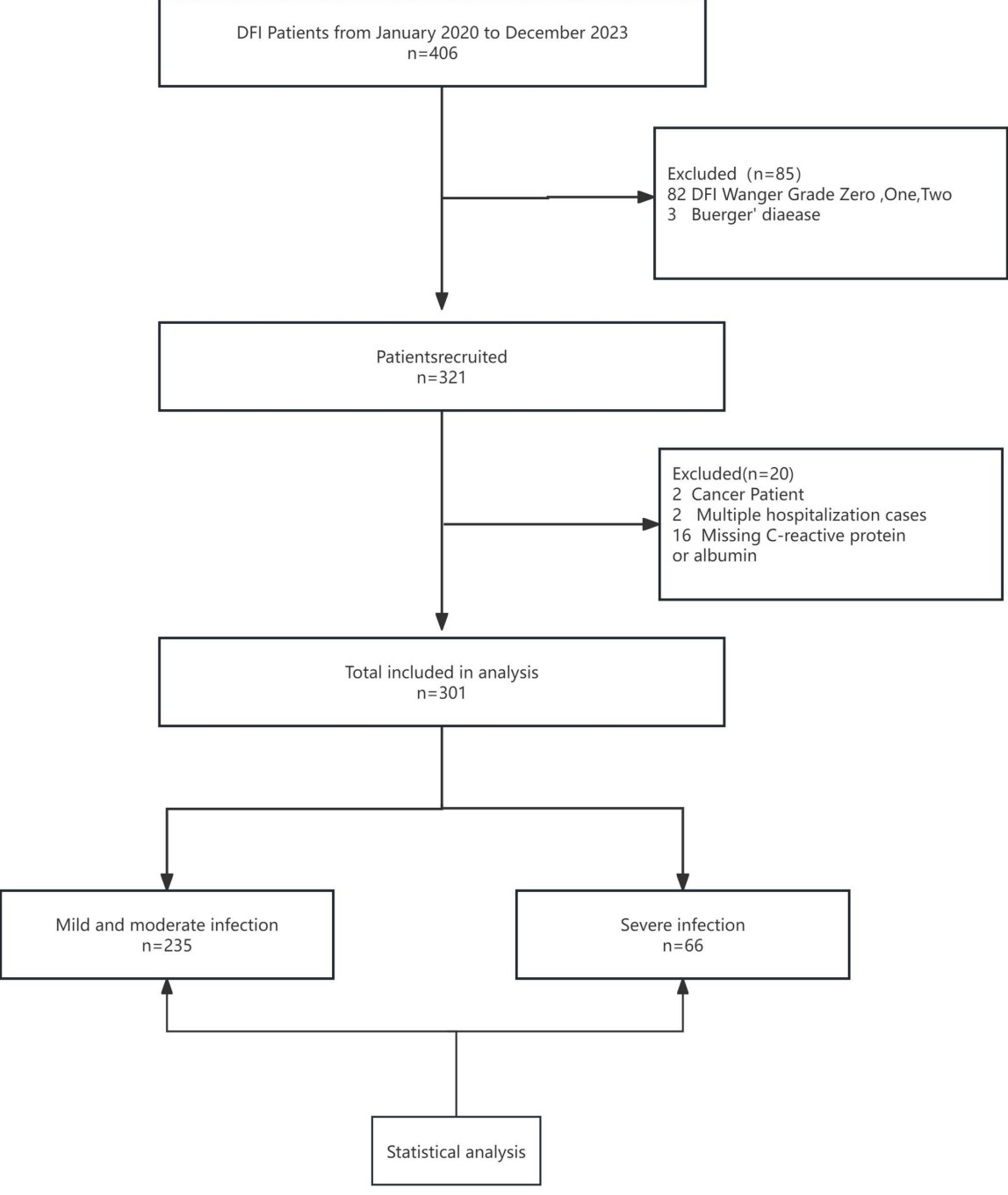

**Fig 1. Flowchart of the patient selection process.**

Both non-adjusted and multivariateadjusted models were used. Variables were adjusted if they changed the odds ratio by at least 10% after being added to the model, and clinically significant variables were also adjusted. Model 1 was adjusted covariates including age and sex, and model 2 adjusted for variables from model 1 plus smoking, diabetes duration, hypertension, coronary heart disease, cerebral infarction, BMI, Mean arterial pressure. A

multivariate-adjusted restricted cubic spline model was constructed to establish the OR curves at 3 knots to examine the possible nonlinear dose response association between CRP and DFI severity.We used a two-piece-wise logistic regression model with smoothing to analyze the association threshold between CRP and DFI severity after adjusting the variables in Model 3. The likelihood-ratio test and the bootstrap resampling method were used to determine inflection points.

In addition, it was determined whether the relationship between CRP and DFI was stable in different population groups. Subgroup analyses were performed according to age, sex, BMI, smoking status, hypertension, coronary heart disease and cerebrovascular disease.

The R statistical software (version 4.2.2, http://www.R-project.org, The R Foundation) and the Free Statistics Analysis Platform Version 1.9 [15] were employed to carry out all the analyses, and p value less than 0.05 represented statistically significance.

## Results

### Baseline characteristics

Baseline characteristics of study participants by CRP category are summarised in Table 1. Of the 406 patients with DFI, 301 eligible individuals were included. The participants comprised 208 male and 93 female, with an average age of 65.2±12.0 years, and 21.9% of the patients had severe DFIs. Differences were noted among the three groups in terms of the severity of DFI, FBG, RBC, hemoglobin, WBC, neutrophil percentage, platelet count, and albumin (all P<0.05). Otherwise, the distribution of patient characteristics was similar (P>0.05).

**Table 1. Baseline Characteristics.**

| Variables | Total | C-reactive protein (mg/L) | | | P |
|---|---|---|---|---|---|
| | | T 1( ≤ 0.62 to ≤ 37.03) | T 2( < 37.03 to ≤ 109.4) | T 3( < 109.4 to ≤ 353.99) | |
| Baseline characteristics | | | | | |
| Sex, n (%) | | | | | 0.655 |
| Male | 208 (69.1) | 72 (72.0) | 66 (66.0) | 70 (69.3) | |
| Female | 93 (30.9) | 28 (28.0) | 34 (34.0) | 31 (30.7) | |
| Age(years) | 65.2±12.0 | 66.1±12.1 | 65.5±11.2 | 64.1±12.6 | 0.472 |
| BMI (kg/m2) | 23.2±3.5 | 23.5±3.2 | 22.8±3.8 | 23.2±3.5 | 0.328 |
| MAP(mmHg) | 59.0±16.9 | 58.8±17.7 | 59.3±16.7 | 58.8±16.3 | 0.97 |
| Smoking statue, n (%) | | | | | 0.578 |
| Current | 60 (19.9) | 20 (20.0) | 19 (19.0) | 21 (20.8) | |
| Never | 107 (35.5) | 41 (41.0) | 31 (31.0) | 35 (34.7) | |
| Former | 134 (44.5) | 39 (39.0) | 50 (50.0) | 45 (44.6) | |
| Diabetes duration (years) | 15.4±9.5 | 14.3±8.3 | 16.4±9.2 | 15.3±10.7 | 0.305 |
| Indicators related to diabetic foot infection | | | | | |
| DFI microbial community, n (%) | | | | | 0.815 |
| G+ bacterial | 100 (34.2) | 32 (33.7) | 30 (30.6) | 38 (38.4) | |
| G- bacterial | 74 (25.3) | 25 (26.3) | 27 (27.6) | 22 (22.2) | |
| Mixed | 118 (40.4) | 38 (40.0) | 41 (41.8) | 39 (39.4) | |
| Severity of diabetic foot infection, n (%) | | | | | < 0.001 |
| Mild and moderate | 235 (78.1) | 98 (98.0) | 81 (81.0) | 56 (55.4) | |
| Severe | 66 (21.9) | 2 (2.0) | 19 (19.0) | 45 (44.6) | |

*(Continued)*

**Table 1.** (Continued)

| Variables | Total | C-reactive protein (mg/L) | | | P |
|---|---|---|---|---|---|
| | | T 1( ≤ 0.62 to ≤ 37.03) | T 2( < 37.03 to ≤ 109.4) | T 3( < 109.4 to ≤ 353.99) | |
| Amputations statue, n (%) | | | | | 0.243 |
| Major | 18 (6.0) | 5 (5.0) | 6 (6.0) | 7 (6.9) | |
| Minor | 208 (69.1) | 63 (63.0) | 69 (69.0) | 76 (75.2) | |
| Without | 75 (24.9) | 32 (32.0) | 25 (25.0) | 18 (17.8) | |
| Comorbidities | | | | | |
| Hypertension, n (%) | | | | | 0.083 |
| Yes | 153 (50.8) | 44 (44.0) | 49 (49.0) | 60 (59.4) | |
| No | 148 (49.2) | 56 (56.0) | 51 (51.0) | 41 (40.6) | |
| Coronary heart disease, n (%) | | | | | 0.846 |
| Yes | 214 (71.1) | 71 (71.0) | 73 (73.0) | 70 (69.3) | |
| No | 87 (28.9) | 29 (29.0) | 27 (27.0) | 31 (30.7) | |
| Cerebral infarction, n (%) | | | | | 0.484 |
| Yes | 205 (68.6) | 70 (70.7) | 71 (71.0) | 64 (64.0) | |
| No | 94 (31.4) | 29 (29.3) | 29 (29.0) | 36 (36.0) | |
| Laboratory examination | | | | | |
| FBG(mmol/L) | 10.4±4.9 | 9.2±4.4 | 10.5±4.9 | 11.6±5.1 | 0.003 |
| HbA1c(%) | 9.0±2.3 | 8.6±2.5 | 9.0±2.1 | 9.3±2.3 | 0.131 |
| RBC(×10$^{12}$/L) | 3.8±0.7 | 4.0±0.8 | 3.8±0.6 | 3.7±0.7 | 0.016 |
| Hemoglobin (g/L) | 112.±22.7 | 120.5±23.4 | 110.5±20.9 | 107.5±22.1 | < 0.001 |
| WBC (×10$^9$/L) | 11.6±5.8 | 8.0±2.7 | 10.8±4.2 | 15.9±6.5 | < 0.001 |
| Neutrophil percentage | 78.5±10.4 | 71.2±10.3 | 78.6±7.7 | 85.7±7.6 | < 0.001 |
| Platelet (×10$^9$/L) | 296±115.9 | 258.6±92.8 | 309.6±117.2 | 320.8±126.3 | < 0.001 |
| Albumin(g/L) | 30.8±6.0 | 34.5±5.3 | 30.2±4.6 | 27.6±5.7 | < 0.001 |

Data presented are mean±SD, median (Q1-Q3), or N (%) DFI, Diabetic foot infections; G+, Gram-positive bacterial; G-, Gram-negative bacterial; BMI, body mass index; MAP, Mean arterial pressure; FBG, Fasting blood glucose; HbA1c %,Glycosylated hemoglobin;RBC, red blood cell counts; WBC, white blood cell count; T, tertiles.

## Association between CRP and DFI severity

In the multivariable logistic regression analysis, the odds ratio (OR) of CRP (per 1mg/dl increase) were consistently significant in all three models (OR:1.01) (Table 2). After adjustment for all covariates, patients with CRP T3 showed a 43.12-fold increase in the DFI severity (OR = 43.12, 95%CI: 9.87~188.25, p<0.001, model 2), compared to CRP T1.

**Table 2. Multivariable logistic regression analyses of CRP and Severity of diabetic foot infection.**

| Variable | OR (95% CI) | | | | | |
|---|---|---|---|---|---|---|
| | Non-adjusted Model | P | Model 1 | P | Model 2 | P |
| CRP(mg/L) | 1.01 (1.01~1.02) | <0.001 | 1.01 (1.01~1.02) | <0.001 | 1.01 (1.01~1.02) | <0.001 |
| CRP tertials | | | | | | |
| T 1(≤0.62 to ≤ 37.03) | 1(Ref) | | 1(Ref) | | 1(Ref) | |
| T2(<37.03 to ≤ 109.4) | 11.49 (2.6~50.82) | 0.001 | 11.92 (2.69~52.83) | 0.001 | 11.89 (2.65~53.24) | 0.001 |
| T3<109.4 to ≤ 353.99) | 39.37 (9.2~168.51) | <0.001 | 40.34 (9.39~173.29) | <0.001 | 43.12 (9.87~188.25) | <0.001 |
| P for Trend | | <0.001 | | <0.001 | | <0.001 |

OR, odds ratio; CI, confidence interval; Ref, reference; T, tertiles. Crude: no other covariates were adjusted.Model1:we adjusted age and gender.Model2:we adjusted gender, age, smoking, diabetes duration, hypertension, coronary heart disease, cerebral infarction, BMI,MAP.

Restricted cubic spline analysis was used to investigate the dose-response relationship between CRP and DFI severity. Fig 2 shows a non-linear relationship between CRP and DFI severity (Fig 2). In the two-piecewise regression models, the adjusted OR for severe diabetic foot infection was 1.039 (95% CI, 1.019–1.06; P < 0.001) in participants with CRP < 105mg/L, whereas there was no association between CRP and DFI severity in participants with CRP ≥ 105mg/L (S1 Table).

## Subgroup analyses

Fig 3 shows the results of the subgroup analyses. In the subgroup analyses, for patients who were at advanced age,BMI and smoking or with coexisting conditions (hypertension, coronary heart disease, and cerebrovascular disease), the associations between CRP and DFI severity were statistically significant, and no significant interaction was detected.

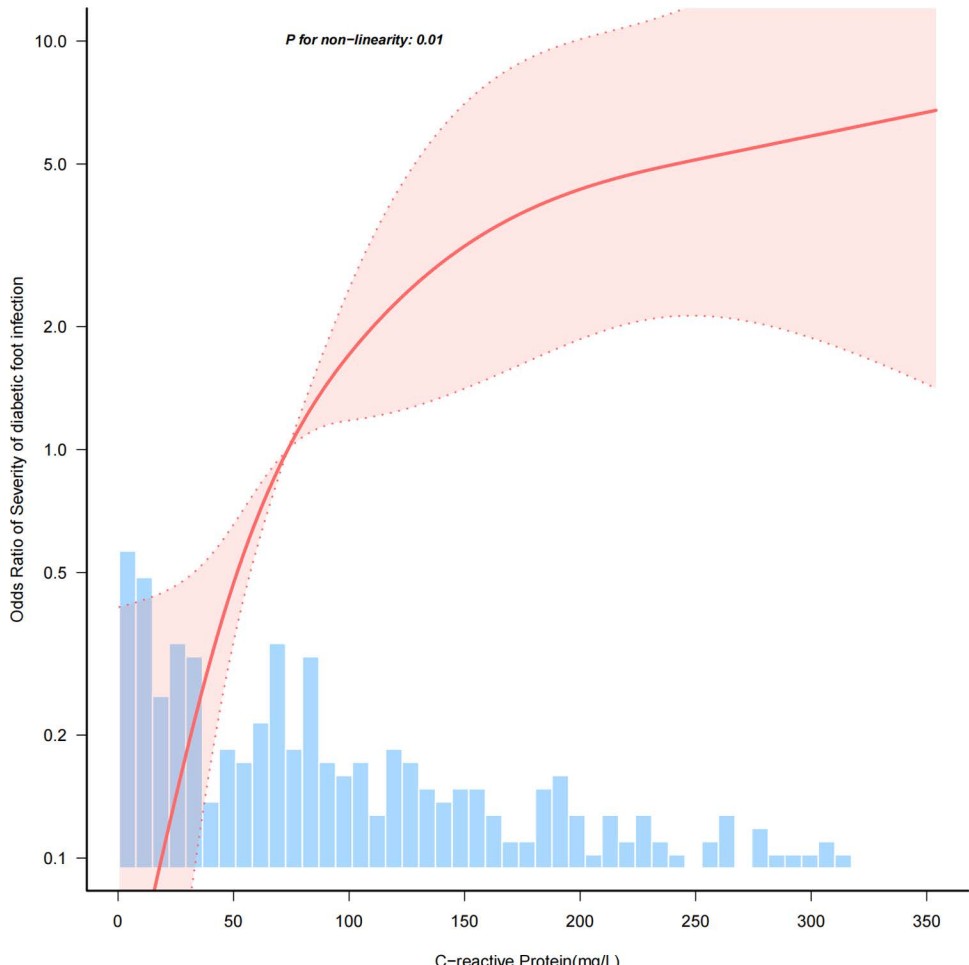

**Fig 2. Nonlinear dose-response relationship between CRP and severity of diabetic foot infection.**

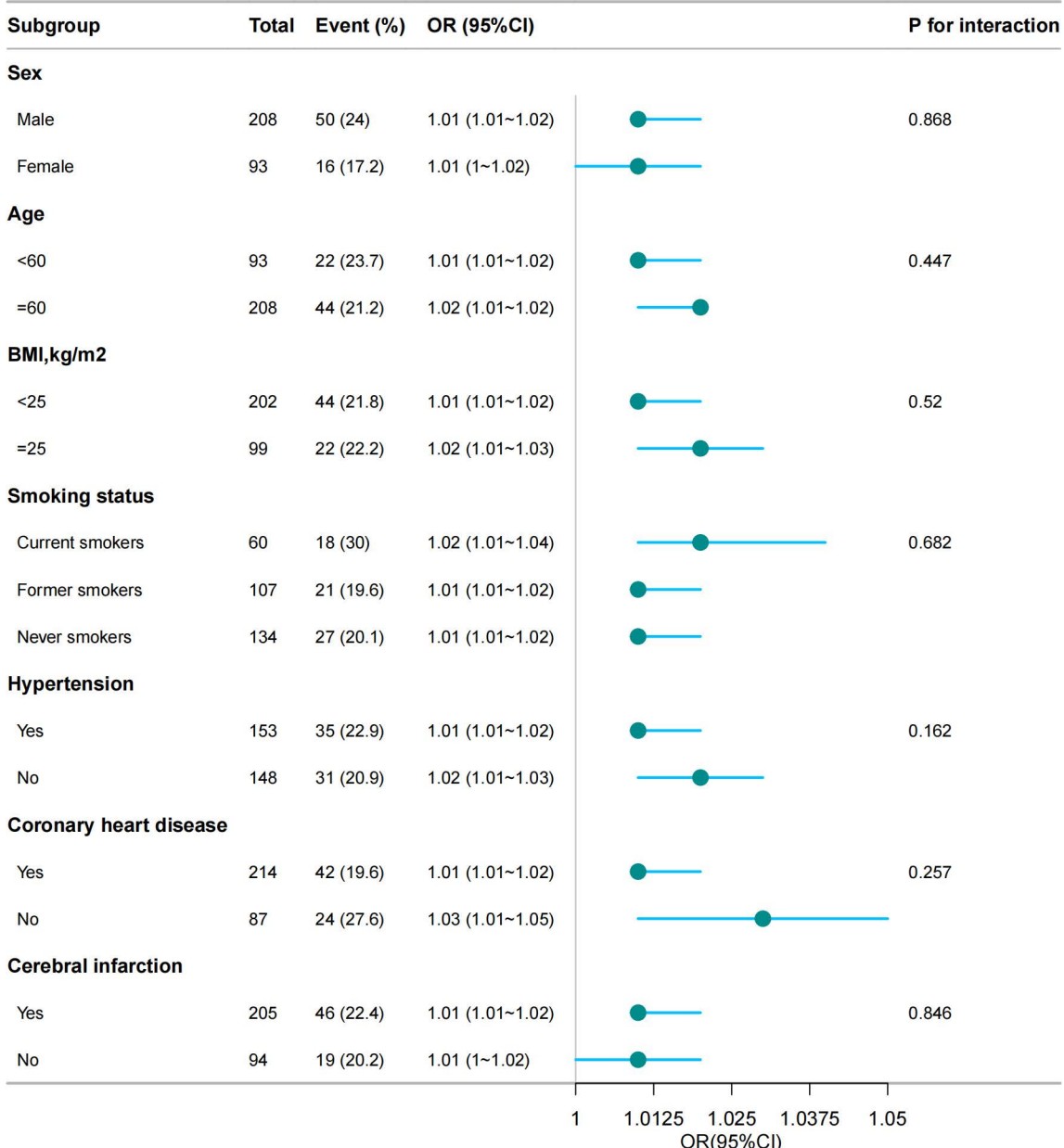

**Fig 3. Stratified analyses of the association between CRP and severity of diabetic foot infection according to baseline characteristics.**

## Discussion

In this retrospective cohort study, CRP was independently associated with DFI severity and demonstrated an inverted L-shaped relationship, with an inflection point of almost 105 mg/L. We observed a positive association between CRP and DFI severity in patients with CRP below 105 mg/L. Both the stratified analyses showed that the relationship between CRP and DFI severity remained robust. To the best of our knowledge, this is the first report of a non-linear association between CRP and DFI severity.

## Comparison with Previous Studies

A multitude of epidemiological research has substantiated the role of CRP associated with the intensity of diabetic foot infections. A study that included a retrospective cohort of 245 patients with DFI found an association between necrosis, serum albumin, erythrocyte sedimentation rate, CRP, and neutrophil-to-lymphocyte ratio values, and severe DFI [10]. Furthermore, In a clinical investigation conducted in South Korean hospitals, which included 123 patients diagnosed with DFI, it was found that higher concentrations of procalcitonin and CRP correlated with increased severity of DFI [16]. Several studies have confirmed the strong relationship between DFI and serum CRP levels [2,10,17,18], but no reports of a non-linear relationship between the two have been found. In our study population, we confirmed a positive correspondence between CRP levels and the severity of DFI. And there is an inverted L-shaped relationship between the two. In other words, serum CRP levels was associated with increased DFI severity when CRP was low than 105 mg/L, and there was no association between the two when CRP was above 105 mg/L.

## Mechanistic insights

It's pertinent to note that CRP is an acute-phase reactant synthesized within hepatocytes [19]. When infection occurs in patients with diabetes-related foot ulcers, plasma levels of CRP rise rapidly and dramatically, up to 1,000 times higher than normal [2,20]. Specific mechanisms may be that CRP is promptly produced by hepatocytes under the regulation of cytokines such as interleukin-6 and interleukin-1, in which upregulation occurs through the stimulation of gene expression by the STAT3, C/EBP, and NF-κB pathways [20].CRP binds to phospholipoproteins of localised bacterial cells in diabetes-related foot ulcers or exerts an immune effect by activating the complement converting enzyme C3. In addition, CRP binds to phagocytes and platelet-activating factors to reduce the inflammatory response [2].The results of this study recommend that CRP is not only a sign of inflammation but is also associated with inflammatory and systemic immune responses [20].

## Clinical implications

Our study found a nonlinear relationship between CRP levels and DFI severity, which was not found in previous studies. In the early stages of severe infections, the immune system can be over-activated, producing large amounts of inflammation-associated factors such as CRP, calcitoninogen and IL-6. However, as the infection progresses, the immune cells may enter a state of exhaustion and the ability to produce inflammatory factors decreases or even reaches saturation, preventing further immune response. Hence, high CRP levels may indicate a more severe infection or complications. These findings highlight the importance of considering low and high CRP levels when assessing DFI severity and tailoring treatment strategies accordingly. Additional research is required to elucidate the fundamental mechanisms behind this non-linear relationship.

The study showed that CRP was not useful to distinguish between noninfected and mild IDFU, and is the most informative marker for differentiating more serious foot infections [17,18]. Our study population of patients with diabetes-related foot ulcers grade 3 and above with moderate to severe infections further confirms that CRP is associated with severe infections, which is consistent with the above findings.

This study had some limitations that need to be noted. First, although we excluded some inflammatory conditions that may affect CRP levels, we cannot exclude the possibility that CRP may be affected by other inflammatory conditions. Second, as with all observational studies, selection bias may have occurred. Thirt,t he study population is from a single institution, which may limit generalisability. Finally,t he wide 95% confidence intervals for the correlation between CRP and DFI severity may be related to sample size, population heterogeneity, and the non-specific nature of CRP as a marker of inflammation, necessitating larger standardised studies with multi-biomarker validation to improve clinical interpretability.

## Concludion

Our study illustrates a nonlinear relationship between CRP levels and DFI severity in patients with DFI, with an inflection point of roughly 105 mg/L. A significant positive correlation was observed between CRP levels and DFI severity at CRP concentrations below 105 mg/L, suggesting the importance of considering the inflammatory status in the early management of DFI. Thus, it is necessary to incorporate CRP levels into diagnostic algorithms for DFI severity, thereby guiding clinicians in developing targeted treatment strategies for patients with diabetic foot infections. Deeper research is required to examine the underlying mechanisms and potential benefical implications of this association.

## Supporting information

**S1 Table. Threshold effect analysis of C-reactive protein on Severity of diabetic foot infection.** Adjustment factors included Sex, Age, Smoking, Diabetes duration, Hypertension, coronary heart disease, Cerebral infarction, BMI, MAP.
(DOCX)

**S1 File. Research datas.**
(XLSX)

## Author contributions

**Conceptualization:** Li Zhang.

**Data curation:** Wenyan Wang.

**Formal analysis:** Xufeng Gao.

**Writing – original draft:** Li Zhang.

**Writing – review & editing:** Meifang He, Yuebin Zhao.

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
