## [Decision Letter · Decision Letter 0]

3 Mar 2025

PONE-D-24-52652Non-linear association between C-reactive protein and severity of diabetic foot infection in patients with diabetes: A retrospective cohort sudyPLOS ONE

Dear Dr. Zhao,

Thank you for submitting your manuscript to PLOS ONE. After careful consideration, we feel that it has merit but does not fully meet PLOS ONE’s publication criteria as it currently stands. Therefore, we invite you to submit a revised version of the manuscript that addresses the points raised during the review process.

We look forward to receiving your revised manuscript.

Kind regards,

Zhanzhan Li

Academic Editor

PLOS ONE

Journal Requirements:

“Taiyuan Bureau of Science and Technology,Science, Technology, and Innovation Program of National Regional Medical Center �202210”

3. In the online submission form, you indicated that [The data underlying this article will be shared by the corresponding author upon reasonable request.].

4.  Please amend your manuscript to include your abstract after the title page.

Additional Editor Comments (if provided):

1.The author should give descriptions about the inclusion and exclusion for study population selection.

2. How do you identify the sample size? You may give some calculation process.

3. For covariates, each covariate should give a specific definition.

4 for laboratory parameters, the examination methods should be descripted.

5. For Table 1, the decimal places should be consistent

6. The study population were divided into three groups according to the C-reactive, how do you identify the cut point? And Is there no population less than 0.62?

7. Some study limitations should be addressed in the discussion.

Reviewers' comments:

Reviewer's Responses to Questions

**Comments to the Author**

1. Is the manuscript technically sound, and do the data support the conclusions?

Reviewer #1: No

Reviewer #2: Yes

Reviewer #3: Partly

2. Has the statistical analysis been performed appropriately and rigorously? 

Reviewer #1: No

Reviewer #2: Yes

Reviewer #3: Yes

3. Have the authors made all data underlying the findings in their manuscript fully available?

Reviewer #1: Yes

Reviewer #2: Yes

Reviewer #3: Yes

4. Is the manuscript presented in an intelligible fashion and written in standard English?

Reviewer #1: No

Reviewer #2: Yes

Reviewer #3: No

5. Review Comments to the Author

Reviewer #1: This study aimed to examine the association between CRP levels and the severity of diabetic foot infections (DFIs) in patients with diabetes using a retrospective "cohort" design and claiming to employ techniques such as generalized additive models and two-piecewise linear regression.

However, the manuscript is laden with extensive reporting flaws and deficiencies, compounded by poor language quality and numerous linguistic errors. These issues render a meaningful scientific review or extraction of valid science from the manuscript in its current state largely impossible. The authors are strongly advised to rewrite the entire manuscript in adherence to an appropriate reporting guideline (e.g., STROBE) and to seek comprehensive language editing by a professional scientific editor specializing in English.

Reviewer #2: General recommendations:

**The title of the article is clear but could be improved by adding a phrase to emphasize its clinical relevance. For example, "Nonlinear Association Between C-Reactive Protein and Severity of Diabetic Foot Infection in Patients with Diabetes: A Retrospective Cohort Study with Clinical Implications" would make the study’s impact more evident.

The abstract is generally well-structured, providing key information on the background, methods, results, and conclusions. However, the introduction could benefit from a more detailed explanation of the need for this review, along with a general overview at the beginning. It could mention the lack of similar studies and their implications for clinical practice. In the methods section, clarify the study period covered and specify the databases searched. In the results section, highlight the strongest associated factors, and the conclusion could include a more specific clinical recommendation, such as the need to integrate CRP levels into diagnostic algorithms for DFI severity.

**In the introduction, there is a good explanation of diabetic foot ulcers (DFUs) as a public health issue. It would be helpful to include more details on the global burden of DFUs and comparisons with other diabetes-related complications. When mentioning previous studies, briefly summarize their findings on CRP and DFI to emphasize the variability in results and the need for this study. The introduction could also more explicitly highlight the novelty of this review compared to existing studies, particularly in exploring the nonlinear relationship between CRP and DFI severity.

**The methods section is thorough and follows standard guidelines for retrospective cohort studies. In the search strategy, consider providing more details on the rationale for choosing specific covariates and how CRP tertiles were determined. The inclusion and exclusion criteria are mostly clear, but you could elaborate on the reasons for excluding patients with other inflammatory conditions or incomplete medical records. It would also be helpful to expand on the statistical rationale for using a two-piecewise linear regression model.

**The results section is well-organized, and the quantitative analysis is robust. However, the data presentation could be more engaging by including a brief narrative summary before diving into the statistical details. For example, introduce the study characteristics and then transition into the statistical results. The figures, while informative, could be improved. Ensure that all figures are presented in high resolution (300 DPI or higher). Additionally, it would be useful to include a detailed explanation of significant findings in a more digestible format, such as simplified explanations of key associations and their implications.

*In the discussion, the findings are effectively related to previous research, but this section could be further strengthened by expanding comparisons with studies from other countries or regions with similar patient populations. Additionally, more explanation is needed on the mechanisms behind the observed threshold effect, such as why CRP levels below 105 mg/L show a different relationship with DFI severity compared to higher levels. Furthermore, the clinical implications of the findings should be expanded. Discuss specific interventions or policy recommendations, such as integrating CRP levels into diagnostic algorithms or developing targeted treatment strategies based on CRP thresholds.

Specific recommendations:

**Introduction

Some sentences are lengthy and could be simplified for better readability. For example:

"Neuropathy and vasculopathy underlie the pathology of diabetic foot ulcers, which in turn are complicated by infections, leading to significant risks such as amputation and even death."

**Materials and Methods

(Lines 35–39)

Grammatical error: "were included the research" should be "were included in the research."

Unclear phrasing: "in1 July 2023 format" is awkward and unclear.

Missing details: It would be helpful to briefly explain the eligibility criteria for inclusion/exclusion.

(Lines 41–46)

Minor grammatical issues: Missing spaces after commas in some places (e.g., "pressure,smoking").

The list of covariates is comprehensive but could be better organized for readability.

(Lines 48–56)

Grammatical errors: "was involved only the skin" should be "involving only the skin."

Inconsistent formatting: "And the ulcers" should not start with "And."

Missing clarity: The criteria for mild, moderate, and severe infections could be presented in a more structured format (e.g., bullet points or subheadings).

(Lines 58–61)

No major issues. This section is clear and concise.

(Lines 63–76)

Grammatical errors: "the demographic traits and clinical metrics was quantified" should be "were quantified."

Repetition: The two-piecewise linear regression model is described twice.

Clarity: The description of the statistical models could be more concise and structured.

**Results

(Lines 78–91)

Grammatical errors: "an average years of 65.2 ± 12.0 years" should be "an average age of 65.2 ± 12.0 years."

Unclear phrasing: "the participants comprised 208 (69.1%) male and 93 female" could be rephrased for clarity.

The sentence about differences among CRP tertiles is lengthy and could be broken into smaller sentences for readability.

Table 1. Baseline Characteristics

The table is well-structured but could benefit from clearer formatting and consistent use of symbols (e.g., "±" for standard deviation).

Some abbreviations (e.g., G+, G-) are not defined in the table legend.

(Lines 93–104)

Grammatical errors: "A high levels of CRP" should be "High levels of CRP."

Repetition: "The link endured" is vague and could be rephrased.

The sentence structure could be improved for clarity.

(Lines 106–112)

Grammatical errors: "After adjustment factors included" should be "After adjusting for factors including."

The description of the nonlinear relationship could be more concise.

(Lines 114–119)

The sentence structure could be improved for clarity.

"Produced uniform findings" is vague and could be rephrased.

**Discussion

The discussion is lengthy and could benefit from subheadings (e.g., "Comparison with Previous Studies," "Mechanistic Insights," "Clinical Implications") to improve readability.

The discussion does not explicitly address the limitations of the study. For example:

The retrospective design may introduce selection bias.

The study population is from a single institution, which may limit generalizability.

CRP levels can be influenced by other inflammatory conditions, which were not accounted for.

**Conclusion

The sentence "This relationship was associated with an increased DFI severity with increasing CRP when CRP was below 105 mg/L" is awkwardly phrased.

I hope my recommendations are helpful to you. Best of luck!

Reviewer #3: The article reports results of a study to identify the association between C-reactive protein and severity of diabetic foot infection. This is a useful study to add to the current evidence on this association, however some areas of methods or text are unclear, as follows:

- Your abstract results report only 301 of the potential sample of 406 patients were included in the analysis. Some information on why over 100 patients were excluded is needed

- Section 2.1 - what are the inclusion and exclusion criteria for 'eligible individuals'? these need to be stated in full. the last sentence of this section needs a grammar edit.

- Section 2.2 - why were other comorbid conditions which may influence CRP levels not included as covariates? e.g. other infections, autoimmune or inflammatory conditions? This needs to be added to a section on limitations of the study

- p.4, paragraph before results - I'd suggest changing the wording of 'effect of CRP' as I don't believe the CRP effects DFI severity, instead it is associated with DFI severity?

- add very wide confidence intervals to the limitations section

- the article needs a language and grammar edit, e.g. first line of introduction 'million grown-up? do you mean adults?

- it is preferred to use the term diabetes-related foot ulcers, rather than diabetic foot ulcer

6. PLOS authors have the option to publish the peer review history of their article (what does this mean? ). If published, this will include your full peer review and any attached files.

**Do you want your identity to be public for this peer review?** For information about this choice, including consent withdrawal, please see our Privacy Policy .

Reviewer #1: No

Reviewer #2: No

Reviewer #3: No

---

## [Editor Report · Decision Letter 1]

14 Apr 2025

Nonlinear Association Between C-Reactive Protein and Severity of Diabetic Foot Infection in Patients with Diabetes: A Retrospective Cohort Study with Clinical Implications

PONE-D-24-52652R1

Dear Dr. Zhao,

We’re pleased to inform you that your manuscript has been judged scientifically suitable for publication and will be formally accepted for publication once it meets all outstanding technical requirements.

Kind regards,

Zhanzhan Li

Academic Editor

PLOS ONE
---

## [Editor Report · Acceptance letter]

PONE-D-24-52652R1

PLOS ONE

Dear Dr. Zhao,

I'm pleased to inform you that your manuscript has been deemed suitable for publication in PLOS ONE. Congratulations! Your manuscript is now being handed over to our production team.

Kind regards,

on behalf of

Dr. Zhanzhan Li

Academic Editor

PLOS ONE